# Modeling Adipogenesis: Current and Future Perspective

**DOI:** 10.3390/cells9102326

**Published:** 2020-10-20

**Authors:** Hisham F. Bahmad, Reem Daouk, Joseph Azar, Jiranuwat Sapudom, Jeremy C. M. Teo, Wassim Abou-Kheir, Mohamed Al-Sayegh

**Affiliations:** 1Department of Anatomy, Cell Biology, and Physiological Sciences, Faculty of Medicine, American University of Beirut, 1107 2260 Beirut, Lebanon; hfbahmad@gmail.com (H.F.B.); rd60@aub.edu.lb (R.D.); jha11@mail.aub.edu (J.A.); 2Laboratory for Immuno Bioengineering Research and Applications, Division of Engineering, New York University Abu Dhabi, 2460 Abu Dhabi, UAE; Jiranuwat.sapudom@nyu.edu; 3Biology Division, New York University Abu Dhabi, 2460 Abu Dhabi, UAE

**Keywords:** adipogenesis, preadipocytes, adipose-derived stem cells, organoids, modeling

## Abstract

Adipose tissue is contemplated as a dynamic organ that plays key roles in the human body. Adipogenesis is the process by which adipocytes develop from adipose-derived stem cells to form the adipose tissue. Adipose-derived stem cells’ differentiation serves well beyond the simple goal of producing new adipocytes. Indeed, with the current immense biotechnological advances, the most critical role of adipose-derived stem cells remains their tremendous potential in the field of regenerative medicine. This review focuses on examining the physiological importance of adipogenesis, the current approaches that are employed to model this tightly controlled phenomenon, and the crucial role of adipogenesis in elucidating the pathophysiology and potential treatment modalities of human diseases. The future of adipogenesis is centered around its crucial role in regenerative and personalized medicine.

## 1. Introduction

### 1.1. What is Adipogenesis?

Adipose tissue is often regarded as a dynamic organ with primordial functions that underline its physiological value. Its versatile contribution to the human body functions include lipid storage, energy homeostasis, and a major share in insulin and other hormonal signaling. Adipose tissue can be classically classified into two different entities: white and brown adipose tissue [1]. Other separate entities also exist, including beige/brite adipose tissue, perivascular adipose tissue, and bone marrow adipose tissue [2]. White adipose tissue represents the largest share of fat that is usually present in the adult human body and is mainly responsible for the aforementioned functions [1]. As a matter of fact, adipokine and cytokine secretion underlines the role of white fat as an endocrine tissue in itself [3]. Brown adipose tissue, on the other hand, is notably abundant in newborns and hibernating mammals. Although adipose tissue encompasses a multitude of cells (macrophages, blood cells, fibroblasts, endothelial cells, and stem cells), mature adipocytes remain the most abundant cell type. It is now well-appreciated that brown and white adipocytes originate from distinct precursor cells. The process by which adipocytes develop from adipose-derived stem cells to form the adipose tissue is called adipogenesis. Adipose-derived stem cells’ differentiation serves well beyond the simple goal of producing new adipocytes. In fact, with the current immense biotechnological advances, the most critical role of adipose-derived stem cells remains their tremendous potential in the field of regenerative and personalized medicine. Herein, we aim to provide a synopsis of the physiological importance of adipogenesis and the current approaches that are employed to model this phenomenon, besides its crucial role in deciphering the mechanisms underlying the pathophysiology and potential treatment modalities of different human diseases.

### 1.2. Studying Adipogenesis to Model Human Diseases

In terms of human diseases, it is worth noting that adipogenesis is not exclusively limited to portraying obesity. In fact, adipogenesis has been employed as a model for a multitude of diseases [4]. When it comes to obesity, it has become a worldwide critical public health burden recently. It has been estimated that, by 2030, 38% of the world’s adults population will be overweight, and 20% of them will be obese [5]. The excess fat mass can be the result of both hypertrophy (increase in cell size) and hyperplasia (increase in cell number) of adipocytes in white adipose tissue [6]. The interplay between the two adipose tissue types plays a key role in regulating obesity. The inflammatory processes in white adipose tissue is a precursor to oxidative stress and the consequent insulin resistance that alters the systemic homeostasis, thus leading to the metabolic syndrome. This is in opposition to brown adipose tissue that is heavily implicated in thermogenesis and energy expenditure. The latter is controlled by the mitochondrial uncoupling protein 1 (UCP-1) [7]. Interestingly, upper-body adiposity is clearly distinct from lower-body adiposity, with the former being a risk factor for obesity and the latter being protective against obesity. Preadipocyte cellular models have been established to further investigate this difference [8]. When it comes to diseases other than obesity, it has been reported that adipose tissue models can be used to study diseases such as cancer and type 2 diabetes mellitus. The impaired insulin signaling forms a tight link between obesity and type 2 diabetes mellitus, making adipocytes a suitable model for the investigation of the disease’s pathophysiology [9]. To note, the isoform-2 of peroxisome proliferator-activated receptor gamma (PPAR-γ2) is one of the major transcription factors that are present in adipose tissue and plays a primordial role in the differentiation process. It was shown to be involved in a variety of metabolic disturbances, such as insulin resistance, dyslipidemia, type 2 diabetes mellitus, and subsequently obesity [10]. Adipogenesis has been also employed to model cancers, such as breast cancer [11,12], prostate cancer [13,14,15], and multiple myeloma [16].

### 1.3. Stem Cells and Adipogenesis

Mesenchymal stem cells are the precursors of adipocytes. These cells differentiate into lipoblasts, then into preadipocytes, and ultimately into the mature adipocytes. Briefly, when adipogenesis takes place, the fibroblast-like preadipocytes differentiate into insulin-responsive adipocytes [17]. The differentiation process is a complex process in which many transcription factors are involved, such as peroxisome proliferator-activated receptor γ (PPARγ), CCAAT/enhancer-binding proteins (C/EBPs), Krüppel-like factor (KLF), and proteins signal transducers and activators of transcription (STATs) [1] (Figure 1). The existence of adipose stem cells is in no way a novel finding: as a matter of fact, it is supported by a large share of evidence. The sole existence of an “adipostat” (fat homeostasis) that is maintained by the pool of adipose stem cells is part of this evidence. Human pathologies like progressive osseous heteroplasia, in which ectopic bone arises from the subcutaneous adipose layer of the skin, also prove the possible “tripotency” of adipose stem cells that can give rise to chondrogenic, osteogenic, and adipogenic cell lines. Finally, the treatment of liposarcomas with ligands that target the previously mentioned PPARγ implies that liposarcomas originate from stem cells, as in the process, liposarcoma cells undergo adipogenesis [18]. Furthermore, the use of adipose-derived stem cells extends beyond the realm of adipocytes alone, as it has been demonstrated that these stem cells can be differentiated into endothelial cells. This further supports the solid crosstalk between adipogenesis and angiogenesis [19]. More importantly, with the current immense biotechnological advances, the most critical role of adipose-derived stem cells remains their tremendous potential in the field of regenerative medicine [20].

### 1.4. Immune Cell Adipocyte Crosstalk

There is considerable clinical evidence that obesity, specifically in combination with type 2 diabetes mellitus (T2DM), causes increased prevalence of a plethora of medical conditions that are immune-mediated. For example, common infections reoccur with higher frequency, exacerbated with increased severity that potentially leads to other complications [21,22]. Higher rates of vaccine failure have also been reported in individuals with obesity [23], perhaps due to compromised adaptive immunity. In all, the association of obesity with a compromised immune system, and a state of chronic low-grade inflammation in adipose tissue [24] is an indication of some degree of cross talk between adipocytes and cells of the immune system. Along with other cell types, such as endothelial and fibroblasts cells, lean adipose tissue contains macrophages for immunologic surveillance purposes. Obese adipose tissue however, can consists of up to 40% of pro-inflammatory macrophages, along with T-cells and B-cells [25,26]. Overall, levels of pro-inflammatory cytokines, TNFα, IL-6, and MCP-1 increased when 3T3-L1 cells were cocultured with murine splenocytes, using a Transwell culture system [26] recapitulating possible cell–cell interaction scenarios in adipose tissues. Interestingly, IL-6 and MCP-1 measured higher when adipocytes and immune cells were in direct contact when activated via lipopolysaccharide (LPS), TNFα measured higher only when exposed to immune-cell-conditioned media. Splenocytes contain a mixed population of cells, while this potentially recapitulates the in vivo situation, the interaction of macrophages and adipocytes alone cannot be elucidated. Using LPS-activated macrophages derived from monocytes purified from human donors, Sarvari et al. [25] showed that IL-6 is macrophage-dependent, as a result of phagocytosis of adipocytes. They also observed lipid droplet accumulation within macrophages after adipocyte–macrophage co-culture [25], which could be a result of the engulfment or uptake of soluble lipids during the phagocytic activity. Garcia-Sabate et al. [27] recently showed that macrophages in 3D mono-culture are able to uptake exogenous low-density lipoprotein and with lipid droplet accumulation dependent on the cell phenotype [27]. Alongside pro-inflammatory cytokines, they also detailed adipokine and growth factor release as a result of lipid accumulation, and, interestingly, levels are dependent not only on macrophage phenotype but on substrate density, as well. Adipocytes are also able to act as antigen-presenting cells, through major histocompatibility complex II, to stimulate IFNγ expressing T-cells [28], which are implicated in autoimmune diseases. This finding is reserved only for large adipocytes of a certain size. The modulation of T-cells correlates with insulin resistance and the involvement of T-cells during obesity is nicely reviewed by Nyambuma et al. 2019 [29]. Despite adipose tissue being known to attract immune cells to proximity, little is known regarding their interactions [25], warranting further research in this field.

### 1.5. Modeling Adipogenesis via 2D and 3D In Vitro Models and In Vivo Animal Models

Modeling the growth of adipocytes in vitro has been extensively studied in the recent years [4] (Figure 1). Notably, the 3T3-L1 cell line, which can differentiate from fibroblasts to adipocytes, remains one of the most frequently used cell lines with standardized and readily available protocols [30]. Two-dimensional (2D) models, however, often fail to precisely replicate the true complexity of adipogenesis. Animal models that are characterized by an extensive lipid deposition in skeletal muscles that is often seen in several human pathologies like myopathies may be considered as acceptable models for studying the mechanisms behind adipogenesis. Wagyu cattle represent a notable example [31]. As much as they are useful, animal models have many drawbacks, including their high cost, their time-consuming isolation procedures, and their failure to recapitulate human pathophysiology due to species differences [32]. Fortunately, it is worth noting that the fidelity in modeling human adipocytes has further improved with the use of human preadipocytes and the previously mentioned adipose-derived stem cells. Add to that the rich interaction of adipocytes with their environment is no longer a secret after the use of 3D models and co-cultures [4,33]. Naturally occurring and biocompatible silk protein scaffolds offered a unique advantage in bioactive adipose tissue engineering [33]. One of the major advancements in culture techniques is the employment of scaffold-free methods in which 3D adipose spheroids are generated from immortal mouse or human pre-adipocyte. Three-dimensional spheroids have been shown to have a more abundant expression and secretion of adiponectin as compared to 2D culture. Their ability to secrete pro-inflammatory cytokines equips them with a superior ability of resisting culture or toxin associated stress. Finally, 3D spheroids that are generated from brown adipose tissue have a higher retention of brown adipose tissue markers than the classical 2D cultures cells from the same origin [32]. It is worth noting here that the superior characteristics of 3D cultures were exploited to model breast cancer, as previously mentioned [12].

## 2. Modeling Adipogenesis via Two-Dimensional (2D) Models

### 2.1. Classical Cells Lines for Studying Adipogenesis

#### 2.1.1. Mouse Cell Lines to Study the Adipogenesis

Most of what we know today about adipogenesis and adipocyte biology comes from studies utilizing in vitro systems. This has led to a dissection of the molecular and cellular events that take place during the transition from undifferentiated preadipocytes, that resemble fibroblasts in their morphology, into mature spherical adipocytes filled with lipid droplets [34]. Primary preadipocytes as well as cell lines that represent different stages of adipocyte development have been used (Table 1). Whereas primary cultures can be used to study depot- or age-dependent adipogenic mechanisms in a manner more representative of in vivo mechanisms, their tedious isolation procedures, short life span in culture and great variability in results have led researchers to resort to more practical models, which are established preadipocyte cell line [34]. The most frequently employed of these cell lines are 3T3-L1 and 3T3-F442A [34]. The 3T3-L1 cell line was derived as a subclone from disaggregated 17- to 19-days old Swiss 3T3 mouse embryos, by selecting cells displaying spontaneous lipid accumulation [35]. The 3T3-F442A cell line is a subclone of 3T3-L1 cells that was generated by serially selecting for cells with the ability to form adipocyte clusters in vitro, and as a result, these cells are generally regarded as a model with a more advanced commitment in the adipose differentiation process than the 3T3-L1 parent strain [36]. Importantly, these clonal cells offer a consistent source of preadipocytes to study since they are homogenous in terms of cellular population and differentiation stage, which allows a homogeneous response to treatments [4]. Follow-up studies showed that treating these cells with insulin, dexamethasone (DEX), and 3-isobutyl-1-methylxanthine (IBMX) in the presence of fetal bovine serum (FBS) potentiated adipogenesis in these cells [4]. Other studies have also reported on additional adipogenic agents, such as rosiglitazone [30] or troglitazone [37], that promoted differentiation over shorter periods than the standard protocol.

Moreover, 3T3-L1 cells have been used extensively to investigate the underlying molecular mechanisms of adipogenesis and to evaluate the potential application of various compounds and nutrients in the treatment of obesity [38]. For example, compounds such as purpurin [39], bergamottin [40] and quercetin [41] were shown to inhibit adipogenic differentiation in 3T3-L1 cells mainly through downregulating the expression of two adipogenic factors, CCAAT enhancer-binding protein alpha (C/EBPα) and peroxisome proliferator activated receptor-gamma (PPARγ). On the other hand, several obesogenic compounds, such as the methanolic Valerian root extract [42], oleic acid [43] and vulpinic acid [44], have also been studied, in order to dissect the mechanisms of action during the differentiation process of 3T3-L1 cells. Furthermore, the biological role of several miRNAs, such as miR-152, in the process of preadipocyte proliferation and differentiation, has also been studied by using 3T3-L1 cells [45].

Regarding the 3T3-F442A cell line, it has been used significantly less than its sister clone in adipogenic differentiation studies, despite minimal differences in the culture protocol between the two cell lines. However, 3T3-F442A cells had an important role in identifying the mechanisms by which growth hormone (GH) acts on lipid accumulation or adipocyte maturation [46]. Moreover, these cells were used to identify small molecules that could induce their differentiation, thus contributing to an additional understanding of some of the transcriptional cascades involved in this process. One of these molecules is staurosporine, a selective serine–threonine kinase inhibitor, which was found to induce adipose differentiation of 3T3-F442A cells through GSK3β activation, thus highlighting the Wnt signaling pathway as a player in adipogenesis [47]. Therefore, 3T3-L1 and 3T3-F442A are well-established cell lines for studying various aspects of adipogenesis in vitro, despite a few setbacks in their culture such as their relatively long adipogenic differentiation time or their difficulty to transfect with siRNAs.

Ob17 cells, derived from adipose precursors present in epididymal fat pads of genetically obese (ob/ob) adult mice are employed less frequently [48]. They represent a later stage in preadipocyte differentiation than 3T3-L1 or 3T3-F442A cells and are characterized by low fatty acid biosynthesis [48]. Their non-embryonic origin makes them confer different responsivities to adipogenic and lipolytic stimuli. For example, they were used in a study showing retinoids as potent adipogenic hormones, rather than inhibitors of preadipocyte differentiation [49].

The OP9 bone-marrow-derived mouse stromal cell line was established from the calvaria of newborn mice genetically deficient in functional macrophage colony-stimulating factor. After only seventy-two hours of adipogenic stimuli, OP9 cells rapidly accumulate triacylglycerol, assume adipocyte morphology, and express adipocyte late marker proteins, including glucose transporter 4 and adiponectin, which makes them a suitable model for high-throughput studies [50]. To this end, they have been used to explore the anti-adipogenic activity of various compounds and nutrients such as quercetin [51], ascorbic acid [52], and transforming growth factor β1 (TGF β1) [52]. Moreover, OP9 cells have also been used to study the role of oxidative stress on the adipogenesis process, particularly fullerene effects on adipogenesis-accompanying oxidative stress and inflammatory changes [53].

C3H10T1/2 is another cell line established in 1973, from 14- to 17-day-old C3H mouse embryonic stem cell precursors. Following treatment with 5′-azacytidine, these fibroblast-like cells can be differentiated into different mesodermal cell types such as adipocytes, chondrocytes, osteoblasts, and myocytes [54,55]. Interestingly, the differentiation process of these cells differs from that of previous cell lines, as commitment to the adipocyte lineage requires bone morphogenetic protein 4 (BMP-4) in addition to the traditional adipocyte differentiation inducers [56]. The main applications of C3H10T1/2 cells in recent years have been focused on investigating the molecular mechanisms associated with adipogenesis and leading to obesity [57]. Specifically, the role of different miRNAs, such as miR-195a, in regulating this process was examined in these cells [58]. Other studies elucidated the role of testosterone in inhibiting differentiation through an androgen receptor-mediated pathway [59]. The most recent paper utilizing C3H10T1/2 cells identified the upregulation of brain-derived neurotrophic factor (BDNF) expression in adipocyte progenitors as a feature leading to age-related dysfunction of visceral white adipose tissue [60].

Mouse embryonic fibroblasts (MEFs) can be isolated after disaggregation of embryos at embryonic day 12–14 and can be differentiated to adipocytes with variable efficiency (10–70%). Established MEF lines allow for more sustained proliferation in vitro and obviate the need to harvest new cells repeatedly from embryos [61]. Unlike other cell culture models, immortalized MEFs cannot differentiate spontaneously when exposed to a hormonal cocktail unless a pro-adipogenic transcription factor such as PPARγ or C/EBPα is introduced [62]. Later studies on MEF differentiation have highlighted the important role of PPARγ specifically in this process, as it was shown that C/EBPα can induce adipogenesis only in its presence, whereas the opposite was not true [63,64]. Recently, MEFs have been used to study genes or transcription factors implicated in the adipogenesis process, especially mechanisms related to obesity. Particularly, the effect of fat mass and obesity-associated (FTO) gene on adipogenesis was elucidated, as it was shown to regulate early events via enhancing the expression of the pro-adipogenic short isoform of RUNX1T1, which enhances adipocyte proliferation [65]. Moreover, a recent study has highlighted the role of nuclear β-actin in modulating C/EBPα during adipogenesis, through a chromatin based mechanism, by utilizing β-actin knockout MEF cell lines [66].

#### 2.1.2. Porcine and Feline Primary Preadipocytes: Better Model for the Study of Adipogenesis Because of Higher Similarity to Human Cells

Due to their high adipogenic capacity, and an adipogenic mode similar to that of human preadipocytes, porcine preadipocytes are recognized as a model system that is superior to rodents in the study of preadipocyte differentiation [67]. Up to 80% of porcine stromal-vascular cells can accumulate lipids in serum-free medium because they are composed of a high proportion of preadipocytes that are able to differentiate [68]. Porcine preadipocytes have been used to study the effects of different effectors and hormones on adipose conversion and metabolism. Insulin added at low concentrations increases preadipocyte differentiation and glucocorticoids such as hydrocortisone enhance this effect [69]. On the contrary, growth hormone decreases adipose conversion by enhancing the secretion of IGFBPs which block the adipogenic action of IGF-I [70]. Other studies that focused on porcine preadipocyte models investigated the effects of different effectors on adipocyte dysfunction and metabolism. Cheng et al. observed that retinol binding protein 4 (RBP-4) significantly suppressed differentiation in porcine preadipocytes by decreasing the activation of insulin signaling pathways [71]. In another study, Pang et al. studied the effects of Akt2 and sirtuin 1 (SIRT-1) on lipogenesis which were mediated through a crosstalk between C/EBPα and PI3K/Akt signaling pathways [72]. Additionally, the role of the autocrine motility factor receptor (AMFR) gene in porcine preadipocyte differentiation was also elucidated [73]. In summary, preadipocyte cells isolated from different animals have been the ideal model for the study of adipogenesis and associated metabolic diseases because they are less costly and easy to isolate with well-established cell-culture-differentiation protocols.

### 2.2. Adipogenesis from Adipose-Derived Stem Cells (ADSCs)

Stem cells are specialized cells that are capable of renewing themselves through cell division and can differentiate into multi-lineage cells. These cells are categorized as embryonic stem cells (ESCs), induced pluripotent stem cells (iPSCs), and adult stem cells. Mesenchymal stem cells (MSCs) are non-hematopoietic multipotent adult stem cells which have the capacity to differentiate into mesodermal lineages such, as osteocytes, adipocytes, and chondrocytes, as well as ectodermal and endodermal lineages [74]. Illustration of adipogenesis and specific marker is depicted in Figure 1. Human MSCs are present in white adipose tissue, in the form of adipose-derived stem cells (ADSCs). They provide a promising future in the field of tissue engineering and regeneration, due to their wide availability in fat tissue, which can be obtained by using surgical procedures such as liposuction. ADSCs have been utilized in studies addressing osteoarthritis, diabetes mellitus and heart disease [75]. Being a subset of MSCs, ADSCs can be differentiated into multiple lineages both in vitro and in vivo. In vitro differentiation is induced by selective medium containing lineage-specific induction factors [76]. This typically consists of DMEM medium with serum, 3-isobutyl-1-methylxanthine, indomethacin, dexamethasone, and insulin [77]. Lipid droplets will start to develop after about one week, with the number of droplets increasing over time. After 12–14 days of differentiation, mature adipocytes are obtained. During this period, extracellular matrix (ECM) proteins, including fibronectin, laminin, and various types of collagen, are expressed by mature and immature cells. At the same time, a type-I collagen network is gradually formed to help ASCs differentiate into mature adipocytes. Gene expression of mature adipocytes is specific, including leptin, aP2, peroxisome proliferator activated receptor-γ2, and glucose transporter type 4 [78]. In comparison with other types of stem cells, ADSCs are more active in autocrine production of some growth factors and cytokines, such as vascular endothelial growth factor-D (VEGF-D), insulin-like growth factor-1 (IGF-1), interleukin-8 (IL-8), interleukin-6 (IL-6), and TGF-β1 [79,80]. For this reason, ADSCs have been clinically used in treatments for inflammatory and autoimmune diseases, such as in trials of graft versus host rejection, Crohn’s disease, and multiple sclerosis [81,82]. The use of these cells as cancer therapies still remains controversial because of their immunomodulatory behavior, which could lead to more growth and a higher metastatic potential [83]. It is worth noting, however, that ADSCs taken from obese individuals have reduced differentiation, immunomodulatory, anti-inflammatory, and metabolic functions [84].

In vitro differentiation of ES cells provides an alternative source of adipocytes for study in tissue culture and offers the possibility to investigate regulation of the first steps of adipose cell development. In particular, ES cells facilitate elucidation of the role of different genes that are involved in adipocyte differentiation [85]. The first morphological observation of adipocyte-like cells derived from ES cells was reported by Field et al.; however, the number of differentiated adipocytes was low [86]. Later on, it was shown that the commitment of ES cells into the adipogenic lineage at a high rate requires treatment of ES cell-derived embryoid bodies (EBs) with retinoic acid (RA) in a specific point in time, and that RA could not be substituted by adipogenic hormones nor by PPAR [87]. A more optimized experimental protocol that is used nowadays is composed of, in addition to RA, adipogenic compounds, such as insulin, triiodothyronine (T3), and thiazolidinedione (TZD), and is PPAR-dependent [88]. These adipocytes were shown to display both lipogenic and lipolytic activities in response to insulin and to ß-adrenergic agonists, respectively, indicating that mature and functional adipocytes are indeed formed from ES cells in vitro [87]. ES cells have been used to study to function of different genes in early stages of adipogenesis. The role of PPARs and C/EBPs in the commitment of stem cells into the adipocyte lineage has been addressed by studying their expression during the determination and the differentiation periods of ES cells. For example, PPARγ and C/EBPβ were shown not to be necessary for the commitment of ES cells into the adipocyte lineage, whereas PPARδ is strongly expressed during the determination phase of ES cells, suggesting that it could act as a master gene involved in the commitment of mesenchymal precursors into the adipocyte lineage [89]. Moreover, the important role of leukemia inhibitory factor receptor (LIF-R) in the development of adipose tissue was elucidated by showing that the capacity of LIF-R null ES cells to undergo adipocyte differentiation was dramatically reduced (from 55–70% to only 5–7%) [90].

Since their generation, patient-specific induced pluripotent stem cells (iPSCs) emerged as an unlimited source of adipocytes for autologous cell-based therapy to treat obesity. The capacity of iPSCs to generate functional adipocytes was first reported by Nakao’s group in 2009 [91]. After 12 days of embryoid body formation and culture with adipogenic differentiation medium for an additional 10 days, the study reported lipid accumulation in approximately 15% of the cells found in half of the embryoid body colonies, which was determined by Oil Red O staining and expression of adipogenesis-related molecules such as C/EBPα, PPARγ2, leptin and aP2 [91]. All later iPSC-derived adipocyte studies have focused on the generation of brown adipose tissue (BAT) because of its promising therapeutic potential for treating human obesity and related metabolic disorders [92]. In 2012, the Cowan group used transgenic induction to overexpress PPARγ2, CEBPβ, and PR domain containing 16 (PRDM16). The resulting adipocytes had a multilocular lipid droplet morphology, high mitochondrial content, and strong cytoplasmic Uncoupling Protein 1 (UCP1) staining, consistent with the formation of brown adipocytes [93]. To address the precise adipogenic precursors necessary for the generation of an expandable reservoir of iPSC-derived white or brown adipocytes, the Dani group identified CD73, CD105, CD90, and CD146 to be expressed on the surface of cells in embryoid body (EB) colonies, which is similar to markers found on ADSCs [94]. They also observed that transient treatment with RA resulted in the development of white adipocytes, whereas its absence led to the formation of brown adipocytes that were enriched in PAX3, supporting the notion that PAX3 may play a role in BAT versus WAT fate determination [94].

In summary, considerable progress has been made toward the development of a cell culture system for stem cell-derived adipocytes that functionally mirror many of the attributes associated with tissue-derived adipocytes in vivo. However, more work needs to be done for the development of new and efficient methods for adipocyte derivation and a deeper characterization of their functional properties, in order to realize their full potential in developing safe and effective cellular therapies for many metabolic disorders.

## 3. Modeling Adipogenesis via Three-Dimensional (3D) Culture, Spheroid, and Organoid Models

Three-dimensional (3D) adipocyte cultures have been developed to better understand the precise mechanisms of adipogenesis in a more physiological setting than standard two-dimensional (2D) cell culture systems (Figure 2A). Whereas such 2D approaches have been highly successful in elucidating the biology of subcutaneous adipocytes, they have been suboptimal for recapitulating the biology of adipocytes from less robust sources, such as visceral adipose tissue. This is because preadipocytes isolated from visceral adipose tissue have impaired function with regard to both proliferation and differentiation in rigid polystyrene-treated plates [98]. Therefore, 3D cell culture methods could serve as an alternative for studying how depot-specific differences influence adipocyte biology.

### 3.1. 2 Spheroidal and Organoid Adipocyte Culture Models

Several other groups have described different methods to the fabrication of 3D adipose cultures [32,99]. White adipose tissue (WAT) has been modeled by adipose spheroids, and this has become more commonplace due to the spheroids’ 3D morphological complexity and suitability for high-throughput screening platforms. In one study, Klingelhutz et al. developed a scaffold-free method to generate 3D adipose spheroids from primary or immortal human or mouse preadipocytes [32]. Upon exposure to differentiation cues, mature spheroids secreted higher levels of adiponectin compared to 2D culture and responded more readily to culture- or toxin-associated stress by secreting pro-inflammatory adipokines. Human ADSCs have also been incorporated into hanging drop cultures, as well as conventional spheroids, and successfully differentiated, raising the possibility of using human primary cells or iPSCs in such systems for screening [99]. Regarding inflammation, a 3D spheroid organization of adipose cells was induced by culturing 3T3-L1 preadipocytes on an elastin-like polyethyleneimine (ELP-PEI)-coated surface. This work investigated the effects of a proinflammatory microenvironment on cellular responses, indicating a more differentiated phenotype in 3D spheroid cultures relative to 2D monolayer analogs. Therefore, 3D adipocyte culture systems offer a platform for elucidating the role of microenvironmental stimuli in effecting key phenotypic responses in various metabolic states [100].

Other approaches sought to utilize an in vitro tissue engineering approach of adipose tissue to mimic native tissue, such that long-term sustainable tissue systems can be developed [101]. In native human adipose tissue, each adipocyte is in close contact with nearby endothelial cells (ECs), which allows for nutrients, oxygen, and different molecules to reach the adipocytes and achieve adipose tissue homeostasis [102]. Therefore, scientists have tried to develop 3D culture systems that are more viable and more representative of structural adipose tissue organization in vivo [103]. One such approach evaluated the incorporation of vasculature into this system [104]. The authors demonstrated the coculture of both undifferentiated and adipocyte-differentiated ADSCs with endothelial cells for two weeks in 3D porous silk fibroin scaffolds, which possess more mechanical integrity and slower degradability than other biomaterials. Confocal microscopy images and histological analyses revealed continuous endothelial lumen formation in both differentiated and undifferentiated cocultures. Differentiated adipose cocultures secreted significantly higher levels of leptin and accumulated more lipids than undifferentiated cocultures [104]. A new study by Aubin et al. described the 3D culture of white adipose tissues reconstructed from their cultured adipose-derived stromal precursor cells [105]. These tissues that consisted of human adipocytes surrounded by stroma, were stable and metabolically active in long-term cultures for at least 11 weeks [105]. Compared to media conditioned by human native fat explants, secretion of major adipokines and growth factors was higher in cultured tissues except for HGF. Moreover, exposure to TNF-α, which is a major proinflammatory cytokine, induced changes in gene expression for adipocyte metabolism-associated mRNAs as well as for genes implicated in NF-κB activation [105]. A more recent study by Muller et al. took adipose tissue organoid culture one step further [106]. They developed their 3D system by using the stromal-vascular fraction of human subcutaneous white adipose tissue, which contains both the adipocyte progenitors and endogenous endothelial cells, thus allowing for recreation of a more native environment for adipocytes. Upon induced differentiation, the formed organoids showed dense vascularization among mature adipocytes with unilocular lipid vacuoles. More importantly, when these organoids were transplanted into immunodeficient mice, they formed integrated chimeric vessels between the endothelial cells within the organoids and the recipient’s circulatory system, thus allowing for long-term maintenance in vivo after transplantation [106].

Overall, adipose organoids represent a more physiological platform to study adipogenesis in vitro under normal and pathological contexts. They are a valuable model to decipher mechanisms involved in obesity and associated diseases and to perform large-scale drug screening. Furthermore, they constitute an interesting model to further study the link between adipogenesis and angiogenesis and to serve as a viable option for vascularized autologous adipose tissue for transplantation.

### 3.2. Adipogenesis in Biomimetic Tissue Models

Adipogenesis is a mechanosensitive process which relies on the stiffness of adipocytes and also the cellular microenvironment [107]. Current research aims to reveal to which extent tissue properties can influence adipogenesis. It has been demonstrated that ADCSs exhibited higher expression of early, mid, and late adipogenic transcription factors on soft polyacrylamide hydrogel [108]. In addition, Zhang et al. showed that substrate stiffness can regulate the balance of osteogenesis and adipogenesis in adipose-derived stem cells [109]. Another report suggested the importance of YAP/TAZ mechanoregulated signaling in adipogenesis in both 2D and 3D poly-ethylene glycol (PEG) based cell culture models [110]. The reports listed above suggested the importance of tissue stiffness in adipogenesis. However, adipogenesis occurred in a 3D fibrous microenvironment, known as extracellular matrix (ECM). The ECM provides mechanical supports and contextual biochemical signaling for cell growth, migration differentiation, and survival. Seo et al. demonstrated obesity-dependent changes in interstitial ECM microstructure and mechanics in genetic and dietary mouse models [111]. To address how ECM affects adipogenesis and adipocyte function, Type I collagen-based hydrogel can be utilized to mimic physiologically relevant adipogenic microenvironments, since it can be modified in terms of biophysical and biochemical properties [112]. Tissue culture plastic coated with type I collagen has been reported to enhance adipogenesis [113], while type VI collagen appears to restrict adipose expansion [114]. Interestingly, by blocking an attachment of integrin alpha 6 to laminin, it allowed differentiation of adipocytes due to sustained anti-adipogenic RhoA activity [115]. A study by Emont et al. established a 3D collagen hydrogel culture system that could differentiate visceral adipocytes as robustly as subcutaneous adipocytes [116]. The study showed improved expression of differentiation and metabolic markers, such as PPAR-γ, TNFα, IL- 6, and adiponectin, as well as morphological characteristics that more closely resembled the depot of origin. For example, subcutaneous adipocytes had higher expression of brown adipocyte-selective markers, and visceral adipocytes had higher expression of inflammatory markers [116]. Another report suggested an enhancement in collagen stiffness via ethylene glycol-bis-succinic acid N-hydroxy succinimide ester (PEGDS) crosslinker could increase adipocyte fibrotic functions [117]. As shown in Figure 2B, the interactions between adipocytes and collagen fibrils could be observed. In addition, Seo et al. reported that collagen fibril diameter can also trigger adipocyte fibrotic functions [118]. Although 3D models were used to study adipocyte functions, less is known about whether tissue microstructure and stiffness, as well as other ECM components, influence adipogenesis. Besides collagen, other materials and bioprinting technology might support the design and study of adipogenesis, for example using methylated or crosslinked gelatin [119,120]. However, they are widely used in translational medicine, rather than in the basic research. In addition to the above, 3D models could also allow for the addition of vasculature and immune components (i.e., macrophages), acting as representative mini-adipose organ structures.

### 3.3. On-A-Chip Technology for Adipogenesis Models

Recently, on-a-chip technology is emerging in many research fields, aiming to mimic pathological and physiological microenvironment at organ level. In the field of adipogenesis, fat-on-a-chip models have been developed [121,122,123]. The models can be integrated with multiplexed immunoassay and imaging technology, allowing high-content analysis of single-cell data. Figure 2C showed an example of white adipose tissue [123]. The fat-on-chip technology allows studies ranging from human eating habits and fat formation towards drug development against obesity.

## 4. Clinical Implications of Adipogenesis Models

### 4.1. From Modeling to Treatment for Different Diseases

As previously described, the potential contribution of adipogenesis in the elucidation of the pathophysiology of multiple diseases via constituting a biologically representative model is not its only outstanding attribute. As a matter of fact, adipogenesis may be implicated in the treatment of various human disease. For instance, adipogenesis models could be utilized for lipidomics studies in various clinically relevant areas, including diabetes [124,125], cardiovascular disease [126,127], prostate cancer [128,129], and psychiatric diseases such as schizophrenia [128].

It is true that the involvement of adipogenesis in the novel and scientifically appealing field of regenerative medicine forms a broad conglomeration of potential treatment strategies [18] (Figure 3). However, one must acknowledge that the therapeutic potential of adipogenesis extends beyond this field. Beige fat was shown to be heavily implicated in energy expenditure. This has been exploited for therapeutic purposes, especially when it comes to obesity: White-to-beige adipocyte conversion is probably one of the main processes that orchestrate this therapy [129]. It is also known that the accumulation of fat in visceral adipose tissue is one of the risk factors for increasing insulin resistance and thus accelerating the progression of type 2 diabetes mellitus. Dysfunctional adipogenesis of omental adipose tissue and high levels of 4-hydroxynonenal (4-HNE) are key players in this phenomenon. The in vitro combination of metformin and insulin was shown to decrease the adipogenesis impairment of preadipocytes that are derived from type 2 diabetes mellitus patients [130]. Therefore, it is possible to conclude that there is an immense potential for novel therapies that revolve around adipogenesis and its regulation (Figure 1).

### 4.2. Screening for Potential Novel Therapies and Therapeutic Targets

The improved understanding of the adipose stem cells physiology and adipogenesis in itself has founded a growing scientific movement that aims to identify potential novel therapies and to bypass the long-standing idea that obesity is always coupled with metabolic disease. The recruitment of new adipocytes through adipogenesis is a major determinant of healthy adipose tissue allocation and remodeling in obesity. This process should be considered as a potential novel therapeutic target. Perhaps targeting the key transcription factors that govern the physiologic adipogenesis, like PPARγ, may open the door to a multitude of possibilities [131]. In fact, it has been shown that PPARγ agonists like the thiazolidinediones anti-diabetic drugs can potentiate the adipogenic capacity of PDGFRβ(+) preadipocytes in adult mice, thus promoting the expansion of healthy visceral white adipose tissue [132]. Additionally, one growth factor receptor that would raise a therapeutic interest is the platelet-derived growth factor receptor isoform α (PDGFRα). Once activated, it triggers a downstream signaling cascade that inhibit the differentiation of adipocyte precursors. Moreover, postnatal mosaic deletion of PDGFRα potentiates adipogenesis, whereas adult deletion enhances the formation of beige adipocytes via a β3-adrenergic receptor activation [133]. The utility of adipocyte recruitment is further supported by the suggestion that strategies aimed at augmenting adipogenesis at the expense of minimizing pathological adipocyte hypertrophy are useful in treating metabolic disease [134]. For instance, obese individuals who do not have diabetes have elevated levels of bone morphogenetic protein 4 (BMP4). This highlights the potential role of BMP signaling as a therapeutic target that can promote healthy adipogenesis and prevent the development of obesity-associated metabolic derangements [135]. Another interesting discovery would be the white adipose tissue endogenous adipogenesis-regulatory cells (Aregs) that have a negative effect on adipose differentiation. Although the literature reveals contradictory findings when it comes to correlating the number of Aregs in a specific depot with its adipogenic potential, Aregs can still hold their position as a promising avenue for obesity treatment [136,137]. It has been also revealed in recent reviews that some non-coding RNAs (ncRNAs) could be implicated in regulating adipogenesis, be it in a positive or negative fashion. They could be also exploited as potential biomarkers to track obesity-associated complications [136]. From another perspective, a notable novel study underlines the potential of exploiting adipogenesis to regenerate soft tissues and to enhance wound repair. Cell-free human adipose liquid extracts are used to trigger angiogenesis and adipogenesis [138]. Finally, another recent promising therapy would be the Korean mistletoe-derived polypeptide viscothionin. In vitro and in vivo experiments have demonstrated that it has an anti-adipogenic effect. Viscothionin was shown to inhibit the differentiation of adipocytes and to minimize the accumulation of intracellular lipids. These effects are mainly orchestrated by the activation of 5′-adenosine monophosphate-activated protein kinase (AMPK) [139]. Interestingly, when compared to simvastatin, the oral administration of viscothionin for three weeks significantly improved serum lipid concentration and reduced body fat content in C57BL/6J mice that were made obese through a high-fat diet [139]. Having mentioned all of these findings, we may conclude that adipogenesis constitutes a very promising niche for therapeutic interventions that target obesity.

## 5. Conclusions and Future Directions

The relatively novel field of regenerative medicine is now considered a research hub in modern medicine. However, the proliferation of this field has many requirements that range from a reliable source of stem cells to biomaterial scaffolds and cytokine growth factors. The advances in improving the fidelity of adipogenesis models have contributed in boosting the output of adipogenesis associated regenerative medicine [18]. The multiple applications of adipogenesis in regenerative therapies range from soft tissue reconstruction to finding a solid link with plastic surgery. Adipose tissue can now be grafted for reconstructive purposes. It is known that autologous fat tissue transfer is commonly used in surgeries, but it is heavily limited by several factors, such as a limited vascularity with large grafts, variable graft survival periods, and dangerously elevated resorption rates. Therefore, tissue engineering remains one of the essential goals that should be aimed for when studying adipogenesis. The previously mentioned culture models may be very useful in enhancing the growth of this field [140]. Grafting is surely not the sole method to exploit the versatility of this tissue. In fact, the “tripotency” of adipose stem cells and ability to selectively control their differentiation to several tissue types may become the focus of therapies in a multitude of fields that are not only limited to reconstructive purposes [20,141].

One final target for the future of adipogenesis would be to focus scientific efforts on attaining a personalized, customized, and patient-defined medicine. The path to attain this goal is already being paved. Characterizing adipose tissue dysfunction by analyzing non-coding microRNAs and shotgun lipidomics profiles may be considered as a legitimate starting point [142]. Investigating the role of long noncoding RNAs in adipogenesis regulation may also direct the current research to the right track [143]. Ultimately, one would place a safe bet on any research effort that tackles the key transcription factors that regulate adipogenesis and the differentiation of adipocytes [144].

## Figures and Tables

**Figure 1 cells-09-02326-f001:**
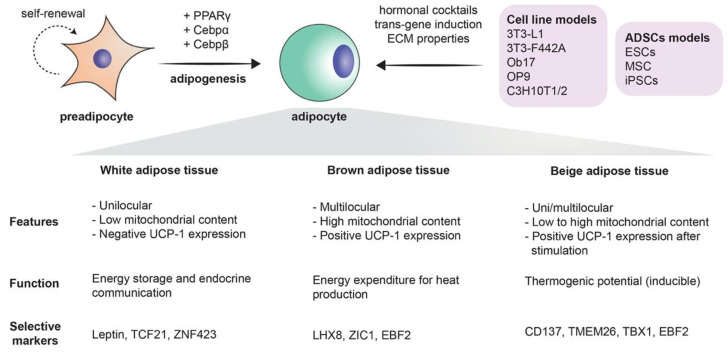
The process of adipogenesis: schematic of undergoing adipogenesis mechanism from immature (preadipocyte) to a mature state (adipocyte). Surface markers used for phenotype characterization of both cells, as well as notable transcription factors that drive adipocyte differentiation, are described. In addition, models utilized to study adipogenesis subcategorized into cell lines and adipose-derived stem cells (ADSCs) with brief description into variable methods used to induce differentiation. Below are featured highlight of each adipose tissue subtypes summarized from other studies [95,96,97].

**Figure 2 cells-09-02326-f002:**
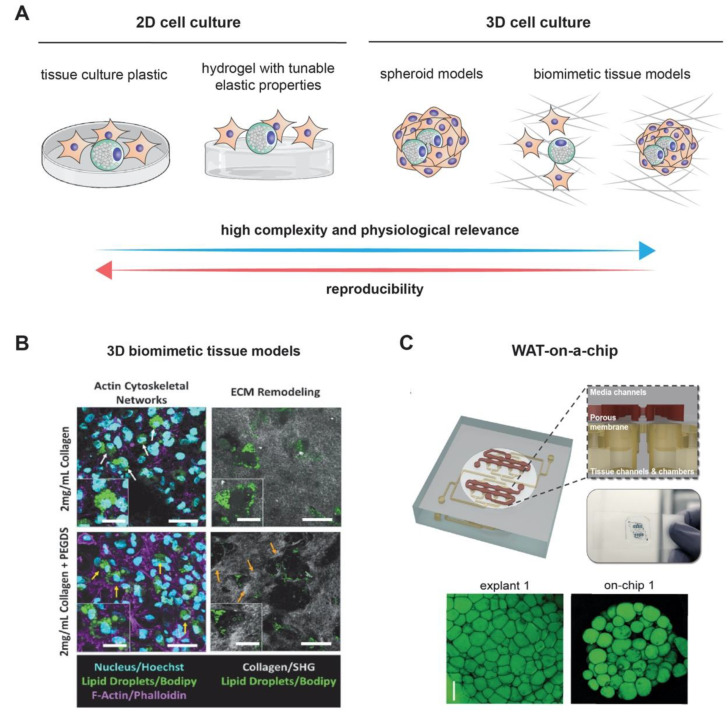
Modeling adipogenesis: (**A**) cell culture models used to study adipogenesis in vitro. Although 2D cell culture models are simple and reproducible, they lack the complexity and physiological relevance exhibited in 3D cell culture models. Example of complex 3D cell culture models, (**B**) using 3D collagen matrices as a biomimetic tissue model and (**C**) a white adipose tissue (WAT)-on-a-chip. Images are adapted with permission from References [99,100], for Figure 2B and 2C, respectively.

**Figure 3 cells-09-02326-f003:**
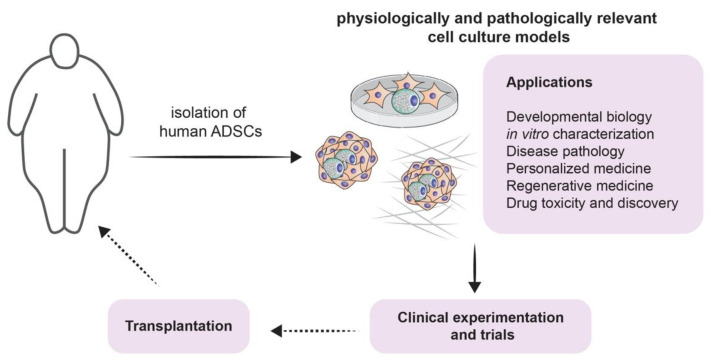
Modeling adipogenesis for experimental approaches and clinical applications: a schematic illustrating potential experimental approaches from isolation of ADSCs throughout culturing, applications (listed), including clinical experiments and trial, and finally towards potential transplantation.

**Table 1 cells-09-02326-t001:** Summary of adipose cell lines discussed in this review.

Cell Lines	Origin	Characteristics
3T3-L1	Disaggregated 17- to 19-days old Swiss 3T3 mouse embryos	Most frequently used preadipocyte model. - Relatively homogenous.
3T3-F442A	Disaggregated 17- to 19-days old Swiss 3T3 mouse embryos	- Similar to 3T3-L1 but more differentiated.
Ob17	Epididymal fat pads of genetically obese (ob/ob) adult mice	- Non-embryonic. - Low fatty acid biosynthesis.
OP9	Calvaria of newborn mice deficient in M-CSF	- Suitable for high-throughput studies.
C3H10T1/2	14- to 17-day-old C3H mouse embryonic stem cell precursors	- Fibroblast-like stem cells. - Suitable for adipogenic commitment studies.
Porcine preadipocytes	Porcine adipose depots	- Resembles more human preadipocytes. - Suitable for the study of metabolic hormones.
Adipose-derived stem cells (ADSCs)	White adipose tissue	- Suitable for adipogenic commitment studies.

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
