# Peer review of "Modeling Adipogenesis: Current and Future Perspective"

_cells, 2020, doi:10.3390/cells9102326_

Round 1

Reviewer 1 Report

The review article entitled “Modelling Adipogenesis: A 2D vs. 3D Perspective” submitted by Hisham F. Bahmad, Reem Daouk, Joseph Azar, Wassim Abou-Kheir and Mohamed Al-Sayegh contains all the key elements and structures and met the criteria and in scope according to the Cells objectives.

The topic is very interesting and topical, especially 3D techniques, as we are witnessing the spread of 3D printing as a technology.

Adipose tissue is also immunologically important, this area was omitted from the introduction, not sufficiently detailed. The introduction is good, but not beautiful, it makes sense to edit from one area to another, there is no diversion, connecting thoughts, so it is not coherent. A little more should be written about brown, beige, and white adipose tissue, as this is a very hot topic in recent years and there are important functional and immunological differences as well. It would be great to see a table of different adipose tissues, their different factors and their function.

I consider it important that animal and human information is well separated: cell lines, differentiation pathways, and so on. Human cell lines are not detailed enough here more information is needed. However, the different differentiation pathways are roughened, there is not enough information e.g. on phenotypic and more important molecular appearance in different precursors and differentiated cells. A chart or table of factors, cytokines, would be good.

AD-MSC and pre-adypocytes can be differentiated not only into white but also beige and brown adipose tissue, this should also be explained.(See examples: PMID: 31251940, PMID: 30967578, PMID: 32316277).

For 3D techniques, a table of the method and the scaffolds used would also be required, as this largely determines the results. 3D printing is underrepresented, this shortcoming needs to be filled. Technologies for 3D printing (FJM, Laser injection, etc.). should also be described because the results available are not the same. This section is missing.

For clinical relevance, I miss the description of pharmacological attack points or what the same and different results are in the different models.

The two figures should be much better, more comprehensive and more informative in view of the comments above. Overall, the topic is good, 2D techniques cell lines are well described, but the description of 3D is rough and much more relevant references are needed.

Overall major revision is needed.

Author Response

Reviewer #1:

The review article entitled “Modelling Adipogenesis: A 2D vs. 3D Perspective” submitted by Hisham F. Bahmad, Reem Daouk, Joseph Azar, Wassim Abou-Kheir and Mohamed Al-Sayegh contains all the key elements and structures and met the criteria and in scope according to the Cells objectives.

The topic is very interesting and topical, especially 3D techniques, as we are witnessing the spread of 3D printing as a technology.

Authors’ Reponse: We thank the reviewer for his/her comprehensive assessment of our manuscript. We modified our manuscript to address all these comments (new changes as tracked changes).

Adipose tissue is also immunologically important, this area was omitted from the introduction, not sufficiently detailed. The introduction is good, but not beautiful, it makes sense to edit from one area to another, there is no diversion, connecting thoughts, so it is not coherent. A little more should be written about brown, beige, and white adipose tissue, as this is a very hot topic in recent years and there are important functional and immunological differences as well.

Authors’ Reponse: We thank the reviewer for his/her comment. Topics related to immune resposes and significage with adipose tissue has now been added. Please refer to section 1.4 “Immune cell adipocyte crosstalk”.

It would be great to see a table of different adipose tissues, their different factors and their function.

Authors’ Reponse: Once again, we thank the reviewer for pointing out this and we totally appreciate that. We agree with the reviewer about adding a table of different adipose tissues, their different factors and their function. A comprehensive table with all the adipocyte feautres, including their subtypes, have been added below the modified figure 1.

I consider it important that animal and human information is well separated: cell lines, differentiation pathways, and so on. Human cell lines are not detailed enough here more information is needed. However, the different differentiation pathways are roughened, there is not enough information e.g. on phenotypic and more important molecular appearance in different precursors and differentiated cells. A chart or table of factors, cytokines, would be good.

Authors’ Reponse: We thank the reviewer for his/her comment. Please refer to Table 1, which highlights cell lines, origin and characteristics of various models used for studing adipogenesis. Cytokines related to adipocyte have been added into figure 1.

AD-MSC and pre-adypocytes can be differentiated not only into white but also beige and brown adipose tissue, this should also be explained.(See examples: PMID: 31251940, PMID: 30967578, PMID: 32316277).

Authors’ Reponse: We thank the reviewer for his/her comment. As mentioned previosuly, a list of features and characteristic of each adipocyte subtype has been added in figure 1. Features of adipocyte subtypes were based on the forementioned studies.

For 3D techniques, a table of the method and the scaffolds used would also be required, as this largely determines the results. 3D printing is underrepresented, this shortcoming needs to be filled. Technologies for 3D printing (FJM, Laser injection, etc.). should also be described because the results available are not the same. This section is missing.

Authors’ Reponse: We thank the reviewer for his/her comment. We have added a new section (refer to section 3.2) “Adipogenesis in biomimetic tissue models”, which highlight various studies that utilizies 3D technologies.

For clinical relevance, I miss the description of pharmacological attack points or what the same and different results are in the different models.

Authors’ Reponse: We thank the reviewer for his/her comment. We have now elaborated more on the clinical relevance of adipogenesis models as well as the pharmacological targets under section “Clinical implications of adipogenesis models”.

The two figures should be much better, more comprehensive and more informative in view of the comments above. Overall, the topic is good, 2D techniques cell lines are well described, but the description of 3D is rough and much more relevant references are needed.

Authors’ Reponse: We thank the reviewer for his/her comment. We have modified figure 2, which comprehensively highlights the difference between the 2D and 3D systems.

Overall major revision is needed.

Authors’ Reponse: We sincerely appreciate the comments raised by the reviewer. We have now addressed all the reviewer’s comments based on his/her suggestions.

Reviewer 2 Report

Authors The manuscript is well organized and  written. In addition, the manuscript is informative in adipose biology research area. It will give a contribution in adipose biology and regenerative medicine. However, authors should be considered the below comments.

  1. Authors should make tables about useful cell lines and 3D spheroid and organoids models for readability. 
  2. Authors should be mention primary cells such as MEF and Stromal vascular fraction (SVF) separately in 2.1
  3.  There is the lacking of information about therapeutic targets in 4.2 section. And I am wondering viscothionin is one of good examples in this concern. 

Author Response

Reviewer #2:

The manuscript is well organized and written. In addition, the manuscript is informative in adipose biology research area. It will give a contribution in adipose biology and regenerative medicine. However, authors should be considered the below comments.

  1. Authors should make tables about useful cell lines and 3D spheroid and organoids models for readability.
  2. Authors should be mention primary cells such as MEF and Stromal vascular fraction (SVF) separately in 2.1
  3. There is the lacking of information about therapeutic targets in 4.2 section. And I am wondering viscothionin is one of good examples in this concern.

Authors’ Reponse: We thank the reviewer for his/her comprehensive assessment of our manuscript. Thank you, Madam/Sir, for pushing us to highly improve the quality and the conclusions of the paper. We really appreciate it and we hope that the replies below will be satisfactory. We modified our manuscript to address all these comments (new changes as tracked changes). As per mentioned:

  1. We thank the reviewer for pointing out this and we totally appreciate that. We agree with the reviewer about adding tables about useful cell lines and 3D spheroid and organoids models for readability. Please refer to the modied figure 1.

  1. We thank the reviewer for his/her comment. We have dedicated a focused section surrounding MEFs within the field as it is being adapted as a good model to study adipogenesis; especially when investigating gene and epigenetics regulation driving adipocyte differentiation.

  1. We thank the reviewer for his/her comment. More information about the therapeutic targets in 4.2 section have been added now as per the reviewer’s comment.

We sincerely appreciate the comments raised by the reviewer. We have now addressed all the reviewer’s comments based on his/her suggestions.

Round 2

Reviewer 1 Report

The authors made the requested changes. The manuscript has improved much in terms of quality and content, presenting the area in a very concise way. The authors have written not only the shortcomings but also complete new chapters that greatly enhance the quality of the manuscript.

A recommend acceptance of this paper,